# The Devil is in the Detail: A Framework for Macroscopic Prediction via Microscopic Models

**Yingxiang Yang**
University of Illinois at Urbana-Champaign
yyang172@illinois.edu

**Negar Kiyavash**
Ecole polytechnique federale de Lausanne
negar.kiyavash@epfl.ch

**Le Song**
Georgia Institute of Technology
lsong@cc.gatech.edu

**Niao He**
UIUC & ETH Zurich
niao.he@inf.ethz.ch

## Abstract

Macroscopic data aggregated from microscopic events are pervasive in machine learning, such as country-level COVID-19 infection statistics based on city-level data. Yet, many existing approaches for predicting macroscopic behavior only use aggregated data, leaving a large amount of fine-grained microscopic information unused. In this paper, we propose a principled optimization framework for macroscopic prediction by fitting microscopic models based on conditional stochastic optimization. The framework leverages both macroscopic and microscopic information, and adapts to individual microscopic models involved in the aggregation. In addition, we propose efficient learning algorithms with convergence guarantees. In our experiments, we show that the proposed learning framework clearly outperforms other plug-in supervised learning approaches in real-world applications, including the prediction of daily infections of COVID-19 and medicare claims.

## 1 Introduction

Many machine learning tasks involve fitting data sets where each data point describes an aggregated effect. For example, country-level COVID-19 infection statistics are aggregation of city-level data; the popularity of a meme in a social network reflects the combined sharing and posting behaviors of the users; a bank's balance sheet numbers depend on the lending and saving practices of all its clients. We refer to the machine learning problems that involve fitting these data sets at the aggregated level as *macroscopic learning*, or macro-learning for short.[1]

**Why learn from macro data?** From an application perspective, performances of many systems are naturally described by statistics at the aggregated level. For example, the GDP reflects the wealth condition of a country, and historic earnings capture the prospect of a company. In these applications, the learning goal is at the aggregated or macroscopic level [1, 14, 33]. Moreover, many applications routinely store and release their data only in the aggregated form, which could be due to privacy concerns [9, 30, 37], or the need to reduce noise in the measurements [34, 35].

**Why learn with micro-models?** Since macro data can often be viewed as aggregated outcomes of fine-grained micro-models, it is natural to learn the underlying micro-models to capture the data-generating dynamics. Additionally, the size of micro data set is often much larger than the macro

data set, providing more samples for training. For example, if we are to learn the monthly sales revenue of a department store, only 12 data points are available per year at the macro-level. However, if we take into consideration that the total sales revenue is the aggregation of thousands of products the store has to offer, then the data set available to the learner is orders of magnitude larger. Furthermore, each micro data point may contain additional features, which can be more easily utilized by micro-models versus macro-models.

**The lack of a micro-macro learning framework.** While access to micro data provides advantage for macro-learning, most existing algorithms are not designed to exploit this symbiotic relationship. Existing approaches essentially fall into two categories:

- **Macro-supervised learning (Macro-SL).** Since the learning objective is at the macro-level, most existing works simply approach the problem with Macro-SL, i.e., supervised learning with macro data [6, 28]. This approach suffers from several short-comings. First, for applications such as time-series forecasting, Macro-SL's performance relies on the design of a good model [20], which requires laborious feature engineering [31]. Second, the size of the macro data set is often small due to aggregation, and Macro-SL does not exploit the micro-level information even when available. Moreover, aggregation can mask the underlying dynamics of the micro data (see Appendix A for an example), and finding a suitable model family to learn the macro data could be hard.
- **Micro-supervised learning (Micro-SL).** Micro-SL approach follows a two-stage procedure: first performing supervised learning on micro data, and subsequently aggregating the micro predictions to obtain a macro prediction [4, 36]. This approach exploits micro information, but suffers from an inductive bias from the choice of Micro-SL objective, leading to large prediction errors after aggregation. Some recent work tries to amend this problem by requiring the macro prediction to follow the desired dynamics, but such dynamics can only be derived for specific model types, e.g., point process models [36] or generalized linear models [7]. Additionally, when the micro data has incomplete labels [7, 8, 9], performing Micro-SL is just infeasible.

**Our Contributions.** This work develops a general macro-learning framework that leverages microscopic models for macroscopic prediction and trains with both macro data and micro data. To achieve this, we propose to directly minimize the loss between macro prediction and macro data, while the macro prediction is estimated through aggregating the microscopic models, as illustrated in Figure 1. More specifically, we formulate the macro-learning problem as an conditional stochastic optimization (CSO), where the learning objective consists of a composition of two expectations, one for modeling the aggregation effect of micro data, and the other for modeling the expected risk over macro data. Minimizing the composite objective can be quite challenging, especially in the nonconvex setting, e.g., when microscopic models involves neural networks. We address this challenge by reformulating the CSO framework as a minimax optimization problem and proposing a novel learning algorithm based on Gradient Descent and Random Search (GDRS). We provide both asymptotic and non-asymptotic convergence analyses of GDRS in the nonconvex-nonconcave setting, which could be of independent interest. Finally, we demonstrate the superiority of the proposed framework over the Macro- and Micro-SL benchmarks in practice.

## 2 Macro-Learning with Conditional Stochastic Optimization (CSO)

### 2.1 Problem Setting

Mathematically, we consider data points represented by a four-tuple: $(\xi, x, y, \bar{y})$, where $\xi, x$ jointly form the feature vector, and $y, \bar{y}$ are the micro and macro labels, respectively. Throughout the paper, we consider data sets of the form $(\xi_i, x_{ij}, y_{ij}, \bar{y}_i)$ for $1 \leq i \leq M$ and $1 \leq j \leq N$, where $(\xi_i, \bar{y}_i)$ corresponds to the macro data and $(x_{ij}, y_{ij})$ corresponds to the micro data. For each pair $(i, j)$, we assume $\xi_i, x_{ij}$ are *i.i.d.* samples from some distribution $\mathbb{P}(\xi, x)$, and together they determine a micro label $y_{ij} \sim \mathbb{P}(y|\xi_i, x_{ij})$. The macro label $\bar{y}_i$ is the observation of the micro labels over the entire population, whose size is potentially much larger than $N$. We refer to $N$ as the aggregation ratio. The goal is to accurately predict the macro labels.

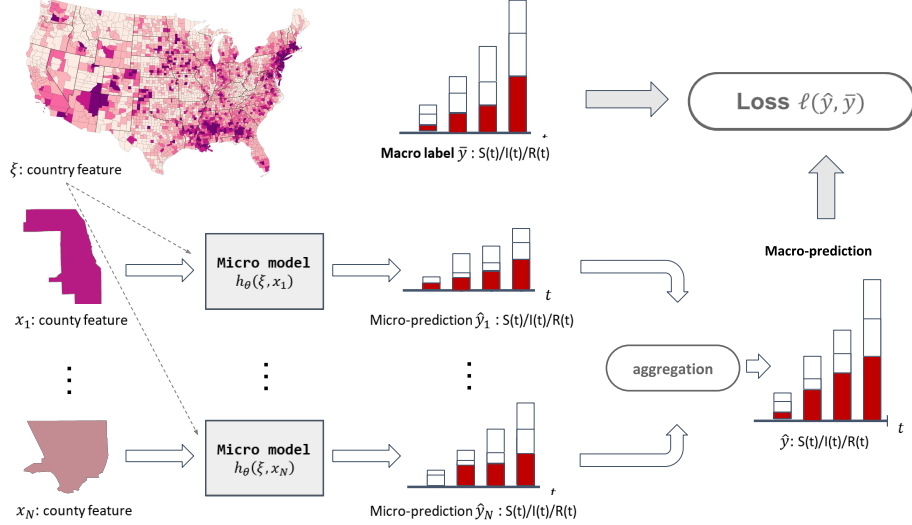

Figure 1: Illustration of macro-learning with micro-models: predict the outbreak of COVID-19 at the country level, by fitting an SIR model [3] for each county. The micro models take in both country-level and county-level features, and output county-level predictions. Those predictions are then aggregated and compared with the country level observation for training.

**Example 1** (COVID-19). *Consider estimating the country-level outbreak of COVID-19 from county-level data. Here, $\xi$ is the country-level feature such as the total population, while $x$ could be the counterpart for each county. The micro label $y$ is a time series documenting the accumulated infections from day 1 to day $T$, and $\bar{y}$ is the number of aggregated infections for the whole country.*

Given the above notation, the Macro-SL and Micro-SL can be formulated as

$$\textbf{Macro-SL:} \quad \min_{\theta \in \Theta_h} \mathbb{E}_{\xi,\bar{y}} \ell(h_\theta(\xi), \bar{y}), \qquad \textbf{Micro-SL:} \quad \min_{\theta \in \Theta_h} \mathbb{E}_{\xi,x,y} \ell(h_\theta(\xi, x), y), \qquad (1)$$

where $h_\theta$ is a parametric model with parameter $\theta \in \Theta_h$. Intuitively, Macro-SL may fall short of good generalization performance when trained with a small amount of macro data. On the other hand, Micro-SL's objective is mismatched to our goal of fitting macro data. To improve Macro-SL by exploiting micro-level information, and at the same time to mitigate the bias within the Micro-SL's training objective, we propose a new optimization formulation to model the macro-learning problem through the so-called conditional stochastic optimization (CSO).

## 2.2 The CSO Formulation

We formulate the macro-learning problem as follows:

$$\textbf{CSO:} \quad \min_{\theta \in \Theta_h} \mathcal{L}(\theta) := \mathbb{E}_{\xi,\bar{y}} \ell \left( \mathbb{E}_{x|\xi}[h_\theta(\xi, x)], \bar{y} \right). \qquad (2)$$

Similar to Micro-SL, CSO adopts a micro-model $h_\theta(\xi, x)$ that takes in both macro and micro features and outputs a prediction of the micro label. Similar to Macro-SL, CSO directly minimizes the loss of macro prediction. In this setting, the learner is not directly supervised by the micro label $y$, but rather through the macro label $\bar{y}$ over the aggregated micro predictions. This aggregate-and-fit procedure indeed matches the learning goal of minimizing the loss at the macro-level, and exploits fine-grained micro features. Furthermore, it allows us to embed $h_\theta$ with a wide spectrum of models (e.g., neural point processes and neural differential equations [10, 24]) compared to existing studies. Note that for generality, we use expected loss and aggregation mean in (2). For finite data sets, the expectations can be replaced by the empirical average and aggregation by the finite-sum structure.

**Example 2** (Formulating COVID-19 with CSO). *Consider forecasting the COVID-19 outbreak in the United States with an SIR model [3]. The data set available contains not only the total number*

*of infections of the entire nation, but also those of each county [16]. More specifically, the data set is $(\xi_i, x_{ij}, \{y_{ij}(t)\}_{t=1}^T, \{\bar{y}_i(t)\}_{t=1}^T)_{i=1,j=1}^{M,N}$ where $\bar{y}_i(t) \in \mathbb{R}^3$ contains the total susceptible, infected and recovered population on day $t$. At the micro level, $x_{ij}$ is the feature (e.g., the population) of county $j$, and $y_{ij}(t) \in \mathbb{R}^3$ is the counter-part of $\bar{y}_i(t)$ for county $j$. Let $SIR_\theta(t|\xi_i, x_{ij}) \in \mathbb{R}^3$ be the prediction of $y_{ij}(t)$ by an SIR model parameterized by $\theta$, then (2) will be specified as*

$$\min_\theta \frac{1}{MT} \sum_{i=1}^M \sum_{t=1}^T \left\| \sum_{j=1}^N SIR_\theta(t|\xi_i, x_{ij}) - \bar{y}_i(t) \right\|^2. \tag{3}$$

*As the aggregation procedure is fixed, (3) can be directly solved using gradient-based optimizers. A similar formulation may be applied to other learning problems, such as predicting aggregated store-level sales using product-level data [4].*

We point out that the CSO formulation has recently been applied in several different contexts, e.g., invariant learning [12], reinforcement learning [12, 13], meta-learning [19], and etc. However, to the best of our knowledge, this framework has not been explored in the context of macro-learning.

An important issue is how to efficiently solve (2). A direct approach is to compute a potentially biased stochastic gradient using the data and perform stochastic gradient descent [19]. This would require a large aggregation ratio, $N$, for each macro data point, in order to ensure small bias in gradient estimation. In addition, [19] showed that such an algorithm requires a total sample complexity of $\mathcal{O}(\epsilon^{-3})$ to achieve an $\epsilon$-optimal solution when the objective is smooth and convex and $\mathcal{O}(\epsilon^{-6})$ to achieve an $\epsilon$-stationary point when the objective is smooth butnonconvex.

In [12], an equivalent minimax optimization formulation was derived to circumvent this bottleneck, allowing to significantly reduce the sample complexity to $\mathcal{O}(\epsilon^{-2})$ for solving CSO formulation in certain cases. This approach has since been applied to reinforcement learning [13], and Bayesian inference [39]. On the other hand, it requires the learner to solve a minimax optimization problem similar to the training procedure of generative adversarial networks (GAN) [18], for which convergence guarantees are only known when the "discriminator" belongs to a reproducing kernel Hilbert space [12]. When neural networks are used to model the "discriminator", no theoretical guarantees are known. Below, we address this challenge by proposing a new algorithm called GDRS to solve CSO, which is proven to be efficient both in theory as well as in practice.

## 3 Provably Efficient Optimization for Macro-Learning

In this section, we first introduce a minimax reformulation of our problem of interest (2), which hinges upon the following interchangeability principle.

**Proposition 1** ([12]). *The CSO formulation in (2) is equivalent to*

$$\min_{\theta \in \Theta_h} \max_{u \in \mathcal{U}} \mathbb{E}_{\xi,x,\bar{y}} h_\theta(\xi, x) u(\xi, \bar{y}) - \ell^*(u(\xi, \bar{y})), \tag{4}$$

*where $\mathcal{U}$ contains all functions on the joint support of $\bar{y}$ and $\xi$, and $\ell^*$ is the Fenchel dual of $\ell$.*

In practice, the dual function $u(\cdot, \cdot)$ is often expressed as a parameterized function, e.g., a linear combination of kernels, or a neural network. In the remainder of the paper, we focus on the parameterized version of (4) evaluated over the training set:

$$\min_{\theta \in \Theta_h} \max_{\lambda \in \Lambda_u} \Phi(\theta, \lambda) := \frac{1}{MN} \sum_{i=1}^M \sum_{j=1}^N \left\{ h_\theta(\xi_i, x_{ij}) u_\lambda(\xi_i, \bar{y}_i) - \ell^*(u_\lambda(\xi_i, \bar{y}_i)) \right\}, \tag{5}$$

where $u(\xi, \bar{y})$ is parameterized by $\lambda \in \Lambda_u$. Denote $\lambda^*(\theta) := \operatorname{argmax}_{\lambda \in \Lambda_u} \Phi(\theta, \lambda)$, $\phi(\theta) := \Phi(\theta, \lambda^*(\theta))$, and $\phi^* := \min_{\theta \in \Theta_h} \phi(\theta)$. By construction of $\Phi$, $\phi(\theta)$ reduces to the CSO objective in (2) when $\operatorname{argmax}_\lambda \Phi(\theta, \lambda) \in \Lambda_u$.

Finally, we point out that when neural networks are used for estimating $h_\theta$ and $u_\lambda$, the above minimax optimization simply becomes nonconvex-nonconcave. Solving such problems efficiently remains largely an open problem. It is known that even widely adopted algorithms such as gradient

---

**Algorithm 1** Gradient Descent with Random Search (GDRS)

---

1: **Input:** $\Phi, \{\eta_k\}_{k=1}^{T-1}, Q, \Theta_h, \Lambda_u, m$.
2: **Initialize:** $\theta^{(1)}, \lambda^{(1)}$.
3: **for** $k$ from 2 to $T$ **do**
4:    Set $\lambda^{(k,0)} = \lambda^{(k-1)}$ and sample $\lambda^{(k,1)}, \ldots, \lambda^{(k,m)}$ iid from $Q$.
5:    Set $\lambda^{(k)} = \lambda^{(k,j)}$ where $j = \text{argmax}_{j' \in \{0,\ldots,m\}} \Phi(\theta^{(k-1)}, \lambda^{(k,j')})$.
6:    Set $\theta^{(k)} = \Pi\left[\theta^{(k-1)} - \eta_{k-1}\nabla_\theta\Phi(\theta^{(k-1)}, \lambda^{(k)})\right]$ where $\Pi$ is the projection operator onto $\Theta_h$.
7: **end for**
8: **Output:** $\theta^{(T)}$.

---

descent ascent (GDA, [18]), can oscillate and diverge except under strong assumptions, e.g., when $\Phi(\theta, \lambda)$ is strongly-concave or satisfies the Polyak-Łojasiewicz condition [25]. In principle, to ensure convergence, the inner maximization problem needs to be computed to high accuracy. When the objective $\Phi(\theta, \lambda)$ is concave or strongly concave in $\lambda$, this can be achieved through iterative gradient-based methods and has been extensively studied in recent literature [25, 32, 22, 23, 26, 38], to name a few. However, when the objective $\Phi(\theta, \lambda)$ is non-concave in $\lambda$, how to properly deal with the inner maximization remains elusive.

### 3.1 Gradient Descent with Random Search

Inspired by the seminar work of Ermol'ev and Gaivoronskii [17], we introduce Algorithm 1, referred to as the gradient descent with random search (GDRS), to solve the aforementioned minimax optimization problem. Algorithm 1 works as follows: at every iteration, parameter $\lambda$ is updated using brute-force search among the current $\lambda$ and a set of $m$ candidates sampled from some distribution $Q$ over the set $\Lambda_u$, and parameter $\theta$ is updated using projected gradient descent based on the best $\lambda$ from the random search. Unlike most other randomized algorithms where the randomness is used to perturb the gradient [27, 29], GDRS directly generates random candidate solutions of $\lambda$ from a pre-selected distribution $Q$, such as the uniform distribution over $\Lambda_u$.

Contrary to the common intuition that one needs to exploit gradients to efficiently find the optimal $\lambda$ for each update of $\theta$, [17] showed that GDRS asymptotically finds an optimal $\lambda^*(\theta)$ and demonstrated the asymptotic convergence of the algorithm when $\Phi(\theta, \lambda)$ is convex in $\theta$. In Section 3.2, we generalize their results by proving asymptotic convergence of GDRS when $\Phi(\theta, \lambda)$ is nonconvex with respect to $\theta$. Furthermore, we establish the first non-asymptotic analysis showing that GDRS converges to an $\epsilon$-approximate stationary point for any fixed $\epsilon > 0$, under proper choices of the sampling size $m$, the step size $\eta_k$, and the number of iteration $T$.

### 3.2 Convergence Analysis

In this section, we provide a theoretical characterization of the asymptotic convergence and finite-time bound of GDRS for solving the minimax optimization (5). Our analysis is quite general and applies to a broad family of nonconvex-nonconcave objectives, not limited to the problem of our interest. Before proceeding, we introduce our assumptions.

**Assumption 1.** *Assume: (i) $\Theta_h$ and $\Lambda_u$ are compact convex sets in Euclidean spaces and $\max_{\theta \in \Theta_h, \lambda \in \Lambda_u} \|\nabla_\theta \Phi(\theta, \lambda)\| = K < \infty$; (ii) $\phi(\theta)$ is L-smooth; (iii) $q(\epsilon) := \min_{\theta \in \Theta_h} q(\epsilon, \theta) > 0$ for all $\epsilon > 0$, where $q(\epsilon, \theta) := \mathbb{P}[\Phi(\theta, \lambda) \geq \phi(\theta) - \epsilon]$ with $\lambda$ randomly sampled from a distribution $Q$; (iv) there exists a constant $\delta_q \geq 0$ such that $q(\epsilon)$ vanishes at the same speed as $\epsilon^{\delta_q}$ as $\epsilon \to 0_+$.[2]*

Assumptions (i) and (ii) are common and standard in minimax optimization. Assumption (iii) implies that at iteration $k$, the probability of finding a $\lambda$ close enough to $\lambda^*(\theta^{(k)})$ is non-zero [17]. This

probability is captured by $q(\epsilon)$, whose asymptotic behavior as $\epsilon \to 0_+$ is assumed to be identical to a polynomial whose order is $\delta_q \geq 0$.

Intuitively, $\delta_q$ can be viewed as a metric that quantifies the "flatness" of $q(\epsilon)$ at 0. The larger the $\delta_q$, the "flatter" $q(\epsilon)$ is at 0. Allowing $\delta_q$ to be non-integer enables us to capture a wider range of limiting behavior of $q(\epsilon)$ at the origin. For example, when $q(\epsilon) = \sqrt{\epsilon}$, we have $\delta_q = 0.5$ while $q'(0_+) = \infty$. In what follows, we characterize the performance of GDRS based on $\delta_q$, however we also note that it is currently unclear to us how large $\delta_q$ would be for complicated distributions, such as a joint distribution over all the neural network's parameters.

Given the nonconvex nature of the problem, the goal is to find an approximate stationary point of $\phi(\theta)$, whose accuracy is often measured through the gradient norm of the Moreau envelop [15]. Define the Moreau envelop of $\phi(\theta)$ as $\phi_\nu(\theta) = \min_{\theta' \in \Theta_h} \{\phi(\theta') + (2\nu)^{-1}\|\theta - \theta'\|^2\}$, where $\nu$ is the proximal parameter. The gradient norm of $\phi_\nu$ measures the distance between two consecutive updates of $\theta$ [5]. We call a solution $\theta$ is an $\epsilon$-approximate stationary point, if $\|\nabla\phi_\nu(\theta)\| \leq \epsilon$. We establish the following convergence results of GDRS.

**Theorem 2** (Convergence of GDRS). *Under Assumption 1, let $\nu = 1/(2L)$.*

(i) *If $\sum_{k\geq 1} \eta_k = +\infty$, and $\sum_{k\geq 1} \eta_k^2 < \infty$, then $\lim_{T\to\infty} \|\nabla\phi_\nu(\theta^{(T)})\| = 0$ in probability. Furthermore, let $\Delta(k, \epsilon)$ be the largest number such that $\sum_{\kappa=k-\Delta(k,\epsilon)}^{k-1} \eta_k \leq \epsilon$. If for all $\epsilon$, $\sum_{k\geq 1}(1 - q(\epsilon))^{m\Delta(k,\epsilon)} < \infty$, GDRS converges almost surely.*

(ii) *Let $\eta_k = \Theta(1/\sqrt{k})$, for any fixed $m \geq 1$,*

$$\min_{1\leq k\leq T} \mathbb{E}\|\nabla\phi_\nu(\theta^{(k)})\|^2 = \mathcal{O}\left(\left(\frac{\log T}{m\sqrt{T}}\right)^{\frac{1}{\delta_q+1}} + \frac{\log T}{\sqrt{T}}\right). \tag{6}$$

*For any fixed $\epsilon > 0$, setting $m = \log_{1-q(\epsilon)} \epsilon$ yields*

$$\min_{1\leq k\leq T} \mathbb{E}\|\nabla\phi_\nu(\theta^{(k)})\|^2 = \mathcal{O}(\epsilon + \log T/\sqrt{T}). \tag{7}$$

The proof can be found in Appendix B. Theorem 2(i) implies that GDRS asymptotically converges to a stationary point. Theorem 2(ii) gives the speed of convergence, which is determined by the critical order of $q(\epsilon)$ at zero. This is rooted in the sampling-based updates of $\lambda$, and depends on the choice of sampling distribution $Q$ and set $\Lambda_u$: the larger $\delta_q$, the flatter it is for $q(\epsilon)$ at 0, and it is harder to obtain a good sample with a fixed sample size $m$. If we allow a large $m$, it is more likely to obtain a sample close enough to the optimal $\lambda^*(\theta^{(k)})$ at iteration $k$. More specifically, if $m = \log_{1-q(\epsilon)} \epsilon$, then GDRS converges to a neighborhood of the stationary point at rate $\mathcal{O}(\log T/\sqrt{T})$, which matches the best-known-rate of projected gradient method for nonconvex minimization [15].

**Remark.** For simplicity of presentation, here we mainly focus on batch GDRS and its convergence. That is, the samples used to compute the gradients are drawn from a training set. Our algorithm and convergence results can be extended to the stochastic variant when the objective and gradient are replaced by their unbiased estimators over a mini-batch of samples at each iteration. We provide detailed discussions in Appendix C. Lastly, we stress that the proposed algorithm and its convergence analysis apply to general minimax optimization problems, e.g., those arising from generative adversarial networks, adversarial machine learning, distributionally robust optimization, etc, which could be of independent interest.

## 4 Numerical Simulations

We present our numerical results comparing GDRS against biased SGD (BSGD), Macro-SL and Micro-SL. In each experiment, we used the same model $h_\theta$ for all algorithms, and tuned the step sizes via grid search. We set $Q$ to be a uniform distribution for each dimension of $\lambda$.

### 4.1 Linear Regression on a Toy Example

**Experiment setting.** Consider the setting where $\xi \in \{-1.0, -0.5, 0.5, 1.0\}$ equiprobably, $x \sim \mathcal{N}(\xi^2, 1)$, $y \sim \mathcal{N}(x + \xi, 1)$ given $x$ and $\xi$, and $\bar{y} \sim \mathcal{N}(\xi + \xi^2, 1)$ given $\xi$. From the distribution

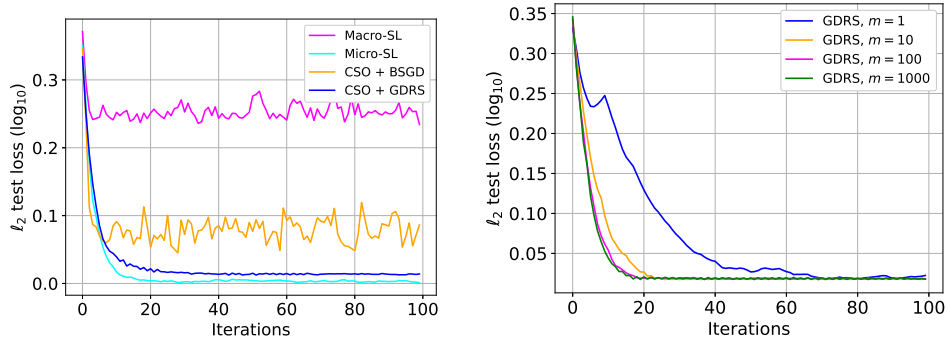

Figure 2: Left: performance comparison of GDRS against benchmarks. Right: performance of GDRS for different $m$.

of $\bar{y}$ we were able to generate training and testing datasets containing 1000 macro data points each. From the distribution of $y$ we were able to generate micro features. We adopted a linear model: $h_\theta(\xi, x) = \theta_1 \xi + \theta_2 x$ with $\theta := [\theta_1, \theta_2]$ restricted within $[-1e5, 1e5]^2$. For Macro-SL, we forced $\theta_2 = 0$ so that the model is only a function of $\xi$. We used a neural network to model $u_\lambda$ with 3 fully connected layers of sizes $[8, 4, 1]$, respectively. When training, we feed the algorithms one macro data point containing 2 micro data points each iteration, and we set $m = 1000$ for GDRS. Finally, we note that since $\xi$ has a discrete alphabet, $u_\lambda$ can also be represented by a dictionary.

**Simulation results.** Figure 2 plots the $\log$ of test error as a function of the number of iterations, averaged over 10 trials. On the left, we plot the performance comparison of GDRS against benchmarks on the test set. We can see that GDRS beats BSGD and Macro-SL. The Micro-SL performs better but requires micro label. On the right, we plot the sensitivity of GDRS's performance over $m$, the number of particles used to find the optimal for $u_\lambda$ for each iteration. We can see that as $m$ increases, the convergence speed of GDRS improves, as also suggested by the convergence analysis.

### 4.2 Learning from Noisy Medicare Data

We studied the macro-learning problem for predicting medicare claims based on a synthetic public use files (SynPUFs) released by the Centers for Medicare and Medicaid Services (CMS)[3].

**Experiment setting.** We simulated a privacy-preserving learning scenario. We divided the entries recorded by the SynPUF dataset into train and test populations, where each entry is a micro data point with 20 features and 9 labels. Among the 20 features, $\xi$ consists of the indicators of 5 most common chronicle conditions such as heart failure and depression, and $x$ consists of the remaining 15 dimensions that contain information such as age and gender. The micro label $y$ is a 3-dimensional label, where each dimension is obtained by summing 3 of the 9 labels that record the annual reimbursement amount associated with the individual under the category of inpatient, outpatient, and carrier. We normalized the data so that the micro label is within $[0, 1]^3$. The macro label is aggregated using all the data sharing the same macro feature within the train or test population.

We trained a fully connected neural network of hidden layer sizes [128, 64] as the micro-model, and a fully connected network of hidden layer sizes [8, 4, 2] for each output dimension of $u_\lambda$. From the noisy training set, we used a mini-batch of size 100 macro data points, each aggregated from 10 micro data points, to compute the gradient and to estimate $\bar{y}$ to supervise the learning. We used Adam optimizer to train the model, and set $m = 10$ for GDRS. We used two loss functions, the $\ell_2$ loss and Csiszár's information divergence [11]: $\text{KL}(x, y) = \sum_i [x_i \log(x_i/y_i) - (x_i - y_i)]$.

**Experiment results.** Figure 3 shows that GDRS significantly outperforms competing algorithms on the test set: the bias resulting from the noise added to the training set as well as the small aggregation ratio cause the performances of these other methods to suffer.

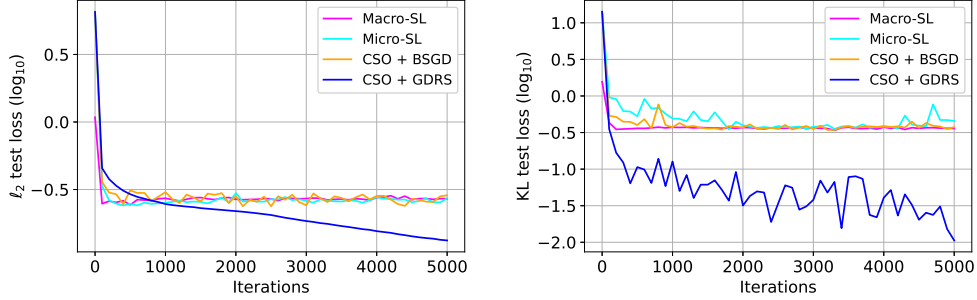

Figure 3: Medicare data: performance of GDRS and benchmarks on SynPUF data set with (i) $\ell_2$ loss (left) and Csiszár's information (right).

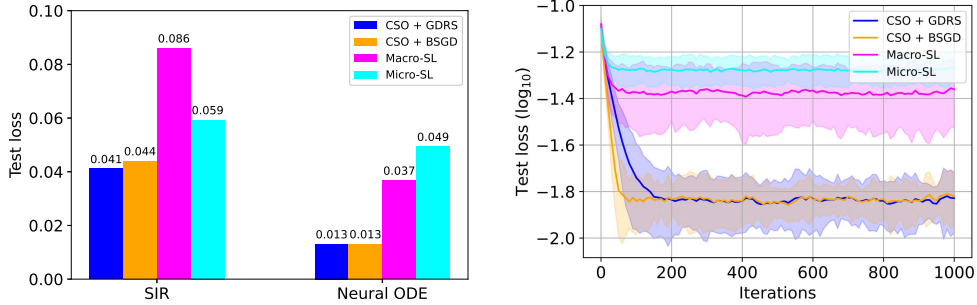

Figure 4: COVID-19 data: comparison between GDRS and benchmarks with SIR/Neural ODE as micro-models (left), and training error trajectory with Neural ODE as the micro-model (right).

### 4.3 Propagation of COVID-19 with SIR Models

Lastly, we applied the CSO framework to learn time series data, by considering the task of predicting the outbreak of COVID-19 across the United States using the data set released by John Hopkins University.[4] The data set contains over 3000 sets of time series, with each set containing $\{S(t), I(t), R(t)\}_{t=1}^{T}$ that document the susceptible, infected and recovered number of people for a county. We used a total of $T = 128$ days of data. Instead of using any specific features for each county, we simply normalized each county's numbers by dividing them with the county's total population. The macro-learning goal was to predict $\{\bar{S}(t), \bar{I}(t), \bar{R}(t)\}_{t=1}^{T}$, obtained by averaging $S(t)$, $I(t)$, $R(t)$ over the training or test set.

**Experiment setting.** We divided the counties randomly into training and test sets with equal probability, and used the SIR model [3] and a Neural ODE [10] to model the underlying dynamics of the disease propagation. The SIR model had only 2 parameters to optimize. For Neural ODE, we used three fully connected neural networks with hidden layer sizes [128, 256] to model $\mathrm{d}S(t)/\mathrm{d}t$, $\mathrm{d}I(t)/\mathrm{d}t$, $\mathrm{d}R(t)/\mathrm{d}t$, respectively. We used the RMSprop optimizer for training, backpropagating through stochastic gradients computed from a mini-batch of size 100 at each iteration. For GDRS, we set $u_\lambda$ to be a vector for the SIR model, and used a fully connected neural network with hidden layer size [16, 8, 4, 2] for Neural ODE, and set $m = 10$.

**Experiment results.** Figure 4 plots the performance of the algorithms on the test data averaged over 10 trials. On the left, we display the average last-iteration performance of each algorithm using different micro-models. On the right, we plot the trajectories of the test error under the Neural ODE model with confidence interval. For SIR model, CSO achieves superior performance compared to Macro-SL and Micro-SL, with slightly better performance if solved by GDRS. The improvement is even more significant for Neural ODE.

## Statement of Broader Impact

This paper takes a preliminary step towards formulating and understanding macroscopic learning, which has been largely ignored until very recently. The quantity of interest is an aggregation of many fine-grained observations, which appear pervasively in modern big data applications, especially those involving privacy considerations. In many cases, one does not have a complete data on the microscopic level, and existing methods that link microscopic data to predict macroscopic labels perform poorly in those cases. We believe that our work makes an important contribution to this emerging field, and sheds light on its wide applicability and potential for further studies from both theoretical and practical aspects.

**Application.** Although we studied concrete examples such as healthcare reimbursement prediction and the prediction of COVID-19, there exists many more learning scenarios where our methods could potentially apply, such as the prediction of social network behaviors or crime activities, predicting total liquidity requirement for financial agencies, just to name a few. Many of these applications are closely related to social sciences and could potentially bring societal impact.

**Theory.** The algorithm and theory developed in this paper apply to general-purposed minimax optimization problems and could benefit a wide spectrum of applications based on minimax optimization, e.g., generative adversarial networks, adversarial machine learning, distributionally robust optimization, safe reinforcement learning, etc. This closely aligns with the community's pursuit of robustness and safety of machine learning.

**Ethics.** There are several ethical questions raised from this work. For example, will supervising the learning of micro-model with macro label effectively preserve privacy? Is the framework more roust to adversarial attacks on the micro or macro labels? We believe that these questions are important in understanding the applicability of macroscopic learning framework but they remain largely unexplored.

## Acknowledgments and Disclosure of Funding

This work was supported in part by ONR grant W911NF-15-1-0479, NSF CCF-1755829, NSF CMMI-1761699, and the Google Faculty award to L.S. The authors would like to thank Pengkun Yang, Junchi Yang, and Yong Lin for useful discussions.

## Footnotes

[1]Throughout the paper, we use macro and micro for macroscopic and microscopic, respectively.

[2]In existing literature, $\delta_q$ is sometimes known as the critical order and appears in generalized Taylor's expansion of $q(\epsilon)$ [2, 21] when $q(\epsilon)$ is fractionally differentiable.

[3]Data source: www.cms.gov.

[4]Data source: `https://github.com/CSSEGISandData/COVID-19`.

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
