[Supplementary Material]

# Appendix

## A   Aggregation Masks Micro Dynamics: an Example

In this section, we provide a simple example where linear aggregation would lose the non-linear relationship embedded within the micro-data. The example is given in Table 1. Other aggregation methods would incur extra tuning, complicating the overall procedure.

| data point | $x_i$ | $y_i$ | intrinsic mapping |
|---|---|---|---|
| $i = 1$ | $(1, 2)$ | 9 | $1 + 2^3 = 9$ |
| $i = 2$ | $(2, 3)$ | 29 | $2 + 3^3 = 29$ |
| aggregated | $(1.5, 2.5)$ | 19 | $1.5 + 2.5^3 \neq 19$ |

Table 1: The mapping from $x_i := (x_{i,1}, x_{i,2})$ to $y_i$ is $y_i = x_{i,1} + x_{i,2}^3$. In this case, linear aggregation distorts the original intrinsic mapping, as $1.5 + 2.5^3 \neq 19$.

## B   Proof of Theorem 2

To prove the theorem, we require the following lemma originally derived in [17]. For completeness, we provide both the statement and the proof.

### B.1   A Technical Lemma

**Lemma 3** ([17]). *For all $\epsilon > 0$,*
$$\mathbb{P}[\phi(\theta^{(k)}) - \Phi(\theta^{(k)}, \lambda^{(k)}) > (2K^2 + 1)\epsilon] \leq (1 - q(\epsilon))^{m\Delta(k,\epsilon)}.$$

**Proof of Lemma 3.** By algorithm construction, $\lambda$ is only updated when the objective value improves. That is, for all $i \in \{0, \dots, m\}$,
$$\Phi(\theta^{(k)}, \lambda^{(k+1)}) \geq \Phi(\theta^{(k)}, \lambda^{(k,i)}). \tag{8}$$
Since $\Phi$ is $K$-Lipschitz continuous with respect to $\theta$, we immediately have
$$|\Phi(\theta^{(k+1)}, \lambda^{(k+1)}) - \Phi(\theta^{(k)}, \lambda^{(k+1)})| \leq K\|\theta^{(k+1)} - \theta^{(k)}\| \leq K^2\eta_k. \tag{9}$$
Likewise,
$$|\Phi(\theta^{(k+1)}, \lambda^{(k,i)}) - \Phi(\theta^{(k)}, \lambda^{(k,i)})| \leq K\|\theta^{(k+1)} - \theta^{(k)}\| \leq K^2\eta_k. \tag{10}$$
Plugging (9) and (10) into (8), we immediately have
$$\Phi(\theta^{(k+1)}, \lambda^{(k+1)}) \geq \Phi(\theta^{(k+1)}, \lambda^{(k,i)}) - 2K^2\eta_k. \tag{11}$$
With the above principle, we can apply the relaxation technique repeatedly, and obtain the following chain of inequalities
$$\Phi(\theta^{(k+1)}, \lambda^{(k+1)}) \geq \Phi(\theta^{(k)}, \lambda^{(k+1)}) - K^2\eta_k \geq \Phi(\theta^{(k)}, \lambda^{(k)}) - K^2\eta_k$$
$$\geq \Phi(\theta^{(k-1)}, \lambda^{(k)}) - K^2(\eta_k + \eta_{k-1}) \geq \Phi(\theta^{(k-1)}, \lambda^{(k-1)}) - K^2(\eta_k + \eta_{k-1})$$
$$\geq \cdots \geq \Phi(\theta^{(k-j)}, \lambda^{(k-j+1)}) - K^2 \sum_{\kappa=k-j}^{k} \eta_\kappa, \tag{12}$$
and
$$\Phi(\theta^{(k-j)}, \lambda^{(k-j+1)}) \geq \Phi(\theta^{(k-j)}, \lambda^{(k-j,i)}) \geq \Phi(\theta^{(k-j+1)}, \lambda^{(k-j,i)}) - K^2\eta_{k-j}$$
$$\geq \Phi(\theta^{(k-j+2)}, \lambda^{(k-j,i)}) - K^2(\eta_{k-j} + \eta_{k-j+1})$$
$$\geq \cdots \geq \Phi(\theta^{(k+1)}, \lambda^{(k-j,i)}) - K^2 \sum_{\kappa=k+1-j}^{k} \eta_\kappa$$
$$\geq \Phi(\theta^{(k+1)}, \lambda^{(k-j,i)}) - K^2 \sum_{\kappa=k-j}^{k} \eta_\kappa. \tag{13}$$

Combining (12) and (13), and setting $j = \Delta(k + 1, \epsilon) - 1$, we get

$$\Phi(\theta^{(k+1)}, \lambda^{(k+1)}) \geq \Phi(\theta^{(k+1)}, \lambda^{(j',i)}) - 2K^2 \sum_{\kappa=k+1-\Delta(k+1,\epsilon)}^{k} \eta_\kappa \tag{14}$$

for all $j' \in \{k - \Delta(k + 1, \epsilon) + 1, \ldots, k\}$ and all $i \in \{0, \ldots, m\}$. By the definition of $\Delta(k + 1, \epsilon)$, we further have

$$\Phi(\theta^{(k+1)}, \lambda^{(k+1)}) \geq \Phi(\theta^{(k+1)}, \lambda^{(j',i)}) - 2K^2\epsilon. \tag{15}$$

Denote set $S_{k+1} = \{\lambda^{(j',i)} : j' \in \{k + 1 - \Delta(k + 1, \epsilon), \ldots, k\}, i \in \{0, \ldots, m\}\}$, we have

$$\mathbb{P}\left[\phi(\theta^{(k+1)}) - \Phi(\theta^{(k+1)}, \lambda^{(k+1)}) \geq (2K^2 + 1)\epsilon\right] \leq \mathbb{P}\left[\phi(\theta^{(k+1)}) - \max_{\lambda \in S_{k+1}} \Phi(\theta^{(k+1)}, \lambda) \geq \epsilon\right]. \tag{16}$$

Now, since the $\lambda$'s in $S$ are drawn i.i.d., we immediately have

$$\mathbb{P}\left[\phi(\theta^{(k+1)}) - \max_{\lambda \in S_{k+1}} \Phi(\theta^{(k+1)}, \lambda) \geq \epsilon\right] \leq (1 - q(\epsilon))^{m\Delta(k+1,\epsilon)}, \tag{17}$$

and hence we have proven the lemma.

∎

Lemma 3 implies that, for any fixed $m$, one can roughly treat the updates of $\theta$'s within the last $\Delta(k, \epsilon)$ iterations as constant when $k$ is large enough, so that the sampled $\lambda$'s have a larger chance of hitting the region where the value of $\Phi(\theta^{(k)}, \cdot)$ is very close to $\phi(\theta^{(k)})$. When the step size vanishes, $\Delta(k, \epsilon) \to \infty$, which directly leads to the following conclusion:

**Corollary 4** ([17]). *When $\eta_k \to 0$ and $\sum_{k \geq 1} \eta_k = \infty$, we have $\Phi(\theta^{(k)}, \lambda^{(k)})$ converges to $\phi(\theta^{(k)})$ in probability.*

In the following, we show that Lemma 3 also has non-asymptotic implications: when $k$ is large, the gradient $\nabla_\theta \Phi(\theta^{(k)}, \lambda^{(k)})$ converges towards the gradient of $\phi(\theta^{(k)})$, and hence leads to the convergence to the stationary point. We develop this idea in more details in the remainder of this section.

## B.2 Proof of (i): Asymptotic Convergence to a Stationary Point

Consider the Moreau envelope of $\phi$:

$$\phi_\nu(\theta) := \min_{\theta'} \left\{ \phi(\theta') + \delta_{\Theta_h}(\theta') + \frac{1}{2\nu}\|\theta - \theta'\|^2 \right\}, \tag{18}$$

where $\delta_{\Theta_h}(\theta') = 0$ if $\theta' \in \Theta_h$ and $\infty$ otherwise. Let

$$\widehat{\theta}^{(k)} = \underset{\theta \in \Theta_h}{\operatorname{argmin}} \left\{ \phi(\theta) + \frac{1}{2\nu}\|\theta - \theta^{(k)}\|^2 \right\}. \tag{19}$$

Then,

$$\begin{aligned}
\phi_\nu(\theta^{(k+1)}) &\leq \phi(\widehat{\theta}^{(k)}) + \frac{1}{2\nu}\|\theta^{(k+1)} - \widehat{\theta}^{(k)}\|^2 \\
&= \phi(\widehat{\theta}^{(k)}) + \frac{1}{2\nu}\left\|\Pi\left[\theta^{(k)} - \eta_k \nabla_\theta \Phi(\theta^{(k)}, \lambda^{(k+1)})\right] - \Pi\left[\widehat{\theta}^{(k)}\right]\right\|^2 \\
&\leq \phi(\theta^{(k)}) + \frac{1}{2\nu}\left\|\theta^{(k)} - \widehat{\theta}^{(k)} - \eta_k \nabla_\theta \Phi(\theta^{(k)}, \lambda^{(k+1)})\right\|^2,
\end{aligned} \tag{20}$$

where the first inequality holds from the definition of the Moreau envelope, and the last inequality holds from the non-expansiveness of the projection operator. Further expanding the last term in (20),

we can further upper bound $\phi_\nu(\theta^{(k+1)})$ by

$$\phi_\nu(\theta^{(k+1)}) \leq \phi(\theta^{(k)}) + \frac{1}{2\nu}\|\theta^{(k)} - \widehat{\theta}^{(k)}\|^2 + \frac{1}{2\nu}\|\eta_k \nabla_\theta \Phi(\theta^{(k)}, \lambda^{(k+1)})\|^2 +$$

$$+ \frac{1}{2\nu} \cdot 2\langle \theta^{(k)} - \widehat{\theta}^{(k)}, -\eta_k \nabla_\theta \Phi(\theta^{(k)}, \lambda^{(k+1)})\rangle$$

$$= \phi_\nu(\theta^{(k)}) + \frac{1}{2\nu}\|\eta_k \nabla_\theta \Phi(\theta^{(k)}, \lambda^{(k+1)})\|^2 + \frac{1}{\nu}\langle \theta^{(k)} - \widehat{\theta}^{(k)}, -\eta_k \nabla_\theta \Phi(\theta^{(k)}, \lambda^{(k+1)})\rangle$$

$$\leq \phi_\nu(\theta^{(k)}) + \frac{1}{2\nu}\eta_k^2 K^2 + \frac{1}{\nu}\langle \theta^{(k)} - \widehat{\theta}^{(k)}, -\eta_k \nabla_\theta \Phi(\theta^{(k)}, \lambda^{(k+1)})\rangle, \tag{21}$$

where the first equality holds by the definition of the Moreau envelope and the construction of $\widehat{\theta}^{(k)}$, and the second inequality holds by upper bounding the gradient norm $\|\nabla_\theta \Phi(\theta^{(k-1)}, \lambda^{(k)})\|^2 \leq K^2$. Now, by the smoothness of $\phi$, we have

$$\phi(\widehat{\theta}^{(k)}) \geq \Phi(\widehat{\theta}^{(k)}, \lambda^{(k+1)})$$

$$\geq \Phi(\theta^{(k)}, \lambda^{(k+1)}) + \langle \nabla_\theta \Phi(\theta^{(k)}, \lambda^{(k+1)}), \widehat{\theta}^{(k)} - \theta^{(k)}\rangle - \frac{L}{2}\|\widehat{\theta}^{(k)} - \theta^{(k)}\|^2. \tag{22}$$

Hence, letting $\nu = 1/2L$, and plugging (22) into (21), we immediately have

$$\phi_{1/2L}(\theta^{(k+1)}) \leq \phi_{1/2L}(\theta^{(k)}) - 2L\eta_k \left[\phi(\theta^{(k)}) - \phi(\widehat{\theta}^{(k)}) - \frac{L}{2}\|\widehat{\theta}^{(k)} - \theta^{(k)}\|^2\right] +$$

$$+ 2L\eta_k \left[\phi(\theta^{(k)}) - \Phi(\theta^{(k)}, \lambda^{(k+1)})\right] + L\eta_k^2 K^2. \tag{23}$$

By the smoothness assumption on $\phi$, it is also a weakly convex function. In particular, $\phi(\theta) + L\|\theta - \theta^{(k)}\|^2$ is $L$-strongly convex. Hence,

$$\phi(\theta^{(k)}) - \phi(\widehat{\theta}^{(k)}) - \frac{L}{2}\|\widehat{\theta}^{(k)} - \theta^{(k)}\|^2 = \left(\phi(\theta^{(k)}) + L\|\theta^{(k)} - \theta^{(k)}\|^2\right) -$$

$$- \left(\phi(\widehat{\theta}^{(k)}) + L\|\widehat{\theta}^{(k)} - \theta^{(k)}\|^2\right) + \frac{L}{2}\|\widehat{\theta}^{(k)} - \theta^{(k)}\|^2$$

$$\geq L\|\widehat{\theta}^{(k)} - \theta^{(k)}\|^2 = \frac{1}{4L}\|\nabla \phi_{1/2L}(\theta^{(k)})\|^2. \tag{24}$$

Hence, plugging (24) into (23) immediately leads to

$$\phi_{1/2L}(\theta^{(k+1)}) \leq \phi_{1/2L}(\theta^{(k)}) - \frac{\eta_k}{2}\|\nabla \phi_{1/2L}(\theta^{(k)})\|^2 +$$

$$+ 2L\eta_k \left[\phi(\theta^{(k)}) - \Phi(\theta^{(k)}, \lambda^{(k+1)})\right] + L\eta_k^2 K^2. \tag{25}$$

Finally, upon telescoping over $k$ from $k_0$ to $\infty$, we have

$$\sum_{k=k_0}^{\infty} \frac{\eta_k}{2}\|\nabla \phi_{1/2L}(\theta^{(k)})\|^2 \leq \phi_{1/2L}(\theta^{(k_0)}) - \min_{\theta \in \Theta_h} \phi_{1/2L}(\theta) +$$

$$+ 2L \sum_{k=k_0}^{\infty} \eta_k \left[\phi(\theta^{(k)}) - \Phi(\theta^{(k)}, \lambda^{(k+1)})\right] + LK^2 \sum_{k=k_0}^{\infty} \eta_k^2. \tag{26}$$

With simple algebraic manipulations, we have

$$\frac{1}{2}\sum_{k=k_0}^{\infty} \frac{\eta_k}{\sum_{t=k_0}^{\infty} \eta_t}\|\nabla \phi_{1/2L}(\theta^{(k)})\|^2 \leq \frac{\phi_{1/2L}(\theta^{(k_0)}) - \min_{\theta \in \Theta_h} \phi_{1/2L}(\theta)}{\sum_{k=k_0}^{\infty} \eta_k} +$$

$$+ 2L\frac{\sum_{k=k_0}^{\infty} \eta_k[\phi(\theta^{(k)}) - \Phi(\theta^{(k)}, \lambda^{(k+1)})]}{\sum_{k=k_0}^{\infty} \eta_k} +$$

$$+ LK^2\frac{\sum_{k=k_0}^{\infty} \eta_k^2}{\sum_{k=k_0}^{\infty} \eta_k}. \tag{27}$$

By the assumption on the step size $\eta_k$, the first and third terms on the right-hand side vanish as $k \to \infty$. By Lemma 3, we have $\Phi(\theta^{(k)}, \lambda^{(k+1)}) \to \phi(\theta^{(k)})$ in probability, which implies that for any $\epsilon > 0$, $\mathbb{P}[\phi(\theta^{(k)}) - \Phi(\theta^{(k)}, \lambda^{(k+1)}) > \epsilon] \to 0$ as $k \to \infty$. Invoking Cesàro's lemma, we further have, for any $\epsilon > 0$, the probability that the second term in (27) being greater than $\epsilon$ is asymptotically

0. Since the left-hand side of (27) is lower-bounded by $\inf_{k \geq k_0} \|\nabla \phi_{1/2L}(\theta^{(k)})\|^2$, we further have $\liminf_{k \to \infty} \|\nabla \phi_{1/2L}(\theta^{(k)})\| = 0$ in probability.

## B.3 Proof of (ii): Vanishing Gradient in the Sense of Mean Square

We start from taking unconditional expectations on (27) over $k \in \{1, \ldots, T\}$, which leads to

$$\frac{1}{2} \mathbb{E}\left[ \sum_{k=1}^{T-1} \frac{\eta_k}{\sum_{t=1}^{T-1} \eta_t} \|\nabla \phi_{1/2L}(\theta^{(k)})\|^2 \right] \leq \frac{\mathbb{E}\phi_{1/2L}(\theta^{(1)}) - \min_{\theta \in \Theta_h} \phi_{1/2L}(\theta)}{\sum_{k=1}^{T-1} \eta_k} +$$

$$+ 2L \frac{\sum_{k=1}^{T-1} \eta_k \mathbb{E}[\phi(\theta^{(k)}) - \Phi(\theta^{(k)}, \lambda^{(k+1)})]}{\sum_{k=1}^{T-1} \eta_k} +$$

$$+ LK^2 \frac{\sum_{k=1}^{T-1} \eta_k^2}{\sum_{k=1}^{T-1} \eta_k}. \tag{28}$$

When $\eta_k = \Theta(1/\sqrt{k})$, the combination of the first and the third terms is upper bounded by $\mathcal{O}(LK^2 \log(T-1)/\sqrt{T-1})$. We now bound the second term.

First notice that we always have

$$\phi(\theta^{(k)}) \geq \Phi(\theta^{(k)}, \lambda^{(k+1)}). \tag{29}$$

Secondly, by Lemma 3 and applying one more round of sampling, we have

$$\mathbb{P}\left[ \phi(\theta^{(k)}) - \Phi(\theta^{(k)}, \lambda^{(k+1)}) \geq (2K^2 + 1)\epsilon \right] \leq (1 - q(\epsilon))^{m(\Delta(k, \epsilon)+1)}. \tag{30}$$

Since $\Theta_h$ and $\Theta_u$ are compact and convex regions, we can assume that there exists a constant $D$ such that

$$\phi(\theta) - \Phi(\theta, \lambda) \leq D. \tag{31}$$

Hence, we can upper bound $\mathbb{E}[\phi(\theta^{(k)}) - \Phi(\theta^{(k)}, \lambda^{(k+1)})]$ with

$$\mathbb{E}[\phi(\theta^{(k)}) - \Phi(\theta^{(k)}, \lambda^{(k+1)})] \leq \epsilon \cdot \mathbb{P}[\phi(\theta^{(k)}) - \Phi(\theta^{(k)}, \lambda^{(k+1)}) \leq \epsilon] +$$

$$+ D \cdot \mathbb{P}[\phi(\theta^{(k)}) - \Phi(\theta^{(k)}, \lambda^{(k+1)}) > \epsilon]$$

$$\leq \epsilon + D(1 - q(\epsilon))^{m(\Delta(k, \epsilon/(2K^2+1))+1)}. \tag{32}$$

Notice that the second term in the right-hand side of (32) is vanishing as $k \to \infty$. Hence, we can also choose a vanishing $\epsilon$ to minimize the upper bound for $\mathbb{E}[\phi(\theta^{(k)}) - \Phi(\theta^{(k)}, \lambda^{(k+1)})]$.

**Finding the optimal $\epsilon$.** We let $\epsilon$ be dependent on $k$, denoted by $\epsilon_k$. By the design of the step size, there exist constants $C_1$ and $C_2$ such that

$$C_1 \epsilon \sqrt{k} \leq \Delta(k, \epsilon_k/(2K^2 + 1)) \leq C_2 \epsilon \sqrt{k}. \tag{33}$$

In order to balance $\epsilon_k$ and $D(1 - q(\epsilon_k))^{m(\Delta(k, \epsilon_k/(2K^2+1))+1)}$, we can fix $C_3$ and $C_4$ to be two positive constants and desire

$$C_3 \epsilon_k \leq D(1 - q(\epsilon_k))^{m(C_2 \epsilon_k \sqrt{k}+1)} \leq D(1 - q(\epsilon_k))^{m(C_1 \epsilon_k \sqrt{k}+1)} \leq C_4 \epsilon_k. \tag{34}$$

Focusing on the right-hand portion of (34), we require

$$\log \frac{D}{C_4 \epsilon_k} \leq -m(C_1 \epsilon_k \sqrt{k} + 1) \log(1 - q(\epsilon_k)) \tag{35}$$

for it to hold. Since $\log(1 - q(\epsilon_k)) \leq -q(\epsilon_k)$, (35) holds true if

$$\log \frac{D}{C_4 \epsilon_k} \leq 2m(C_1 \epsilon_k \sqrt{k} + 1) q(\epsilon_k). \tag{36}$$

Reparameterizing $\epsilon_k$ by $\gamma_k/\sqrt{k}$, we see that (36) holds if

$$\log \frac{D\sqrt{k}}{C_4 \gamma_k} \leq 2m(C_1 \gamma_k + 1) q(\gamma_k/\sqrt{k}), \tag{37}$$

or equivalently

$$\frac{\log \frac{D\sqrt{k}}{C_4 \gamma_k}}{2m(C_1 \gamma_k + 1)} \leq q(\gamma_k/\sqrt{k}). \tag{38}$$

By assumption, there exists a constant $C_5$ such that

$$q(\gamma_k/\sqrt{k}) \geq C_5 \left(\frac{\gamma_k}{\sqrt{k}}\right)^{\delta_q}. \tag{39}$$

Hence, (38) holds true if

$$\frac{\log \frac{D\sqrt{k}}{C_4\gamma_k}}{2m(C_1\gamma_k + 1)} \leq C_5 \left(\frac{\gamma_k}{\sqrt{k}}\right)^{\delta_q}, \tag{40}$$

which holds if

$$\gamma_k = \Omega\left(m^{-\frac{1}{\delta_q+1}} \sqrt{k}^{\frac{\delta_1}{\delta_k+1}} \log^{\frac{1}{\delta_q+1}} k\right),$$

or equivalently

$$\epsilon = \Omega\left(\left(\frac{\log k}{m\sqrt{k}}\right)^{\frac{1}{\delta_q+1}}\right). \tag{41}$$

Hence choosing $\epsilon = \Theta\left(\left(\frac{\log k}{m\sqrt{k}}\right)^{\frac{1}{\delta_q+1}}\right)$ yields

$$\mathbb{E}[\phi(\theta^{(k)}) - \Phi(\theta^{(k)}, \lambda^{(k+1)})] = \mathcal{O}\left(\left(\frac{\log k}{m\sqrt{k}}\right)^{\frac{1}{\delta_q+1}}\right). \tag{42}$$

Noticing that $\eta_k = \Theta(1/\sqrt{k})$, we further have

$$\frac{\sum_{k=1}^{T-1} \eta_k \mathbb{E}[\phi(\theta^{(k)}) - \Phi(\theta^{(k)}, \lambda^{(k+1)})]}{\sum_{k=1}^{T-1} \eta_k} = \mathcal{O}\left(\frac{\sum_{k=1}^{T-1} \frac{1}{\sqrt{k}} \left(\frac{\log k}{m\sqrt{k}}\right)^{\frac{1}{\delta_q+1}}}{\sqrt{T-1}}\right)$$

$$= \mathcal{O}\left(\frac{\log^{\frac{1}{\delta_q+1}}(T-1)}{m^{\frac{1}{\delta_q+1}}\sqrt{T-1}} \sum_{k=1}^{T-1} \frac{1}{\sqrt{k}^{\frac{1}{\delta_q+1}}}\right)$$

$$= \mathcal{O}\left(\left(\frac{\log(T-1)}{m\sqrt{T-1}}\right)^{\frac{1}{\delta_q+1}}\right), \tag{43}$$

which completes the proof.

Meanwhile, if we choose $m = \log_{1-q(\epsilon)} \epsilon$, then the right-hand side of (32) can be further upper bounded by $(1 + D)\epsilon$. Hence, the convergence rate of $\min_{1 \leq k \leq T} \|\nabla \phi_{1/2L}(\theta^{(T)})\|$ is $\mathcal{O}(\epsilon + \log T/\sqrt{T})$.

## C GDRS for Stochastic Objective and its Convergence Rate

We consider when $\Phi$ in Algorithm 1 can only be estimated via $\widehat{\Phi}_{N_k}$, an average of $N_k$ noisy observations of $\Phi$ at the $k$-th iteration. In particular, we define
$$\widehat{\Phi}(\theta, \lambda) = \Phi(\theta, \lambda) + w(\theta, \lambda), \tag{44}$$
to be a random observation, with $w$ being the noise component, and define
$$\widehat{\Phi}_{N_k}(\theta, \lambda) = N_k^{-1} \sum_{i=1}^{N_k} \widehat{\Phi}^{(i)}(\theta, \lambda), \tag{45}$$
where $\widehat{\Phi}^{(i)}$'s are i.i.d. observations of $\widehat{\Phi}$. In this setting, we have the following results.

**Theorem 5** (Convergence of GDRS under noisy observations of $\Phi$). *Under Assumption 1, suppose $\mathbb{E}\|w(\theta, \lambda)\|^2 < \sigma^2 < \infty$. Assume $\sup_{\theta \in \Theta_h, \lambda \in \Theta_u} \|\nabla_\theta \widehat{\Phi}(\theta, \lambda)\|^2 = \widehat{K} < \infty$, $\mathbb{E}[w(\theta, \lambda)] = 0$, and $\mathbb{E}[\nabla_\theta \widehat{\Phi}(\theta, \lambda)] = \nabla_\theta \Phi(\theta, \lambda)$. Let $\{N_k\}_{k \geq 1}$ be an increasing and diverging sequence, and set $m = 1$ in Algorithm 1. Then,*

- *if $\sum_{k \geq 1} \eta_k = +\infty$ and $\sum_{k \geq 1} \eta_k^2 < \infty$, we have $\lim_{T \to \infty} \|\nabla \phi_{1/2L}(\theta^{(T)})\| = 0$ in probability, and almost surely if $\exists \delta > 1$ such that $k^\delta/N_k \to 0$.*

- if $\eta_k = \Theta(1/\sqrt{k})$, and $N_k = \Omega\left(\frac{\sigma^2 k^{\frac{1}{2}(1+\frac{3}{\delta_q+1})}}{\log^{\frac{3}{\delta_q+1}} k}\right)$, then

$$\min_{1 \leq k \leq T} \mathbb{E}\|\nabla\phi_{1/2L}(\theta^{(k)})\|^2 = \mathcal{O}\left(\left(\frac{\log T}{m\sqrt{T}}\right)^{\frac{1}{\delta_q+1}} + \frac{\log T}{\sqrt{T}}\right). \tag{46}$$

**Remark 6.** *Similar to Theorem 2, Theorem 5(i) provides asymptotic convergence of GDRS under stochastic observations of the objective, while Theorem 5(ii) provides convergence rate. Interestingly, one does not need too many samples in order to accurately evaluate $\Phi$. In particular, the larger the $\delta_q$, the less samples one needs at each iteration. This is because the random search procedure eventually dominates the speed of convergence. As a direct consequence, setting $N_k = \Omega(k^2)$ suffices to guarantee the same rate as if we are observing noiseless $\Phi$.*

To prove the theorem, we require the following lemma originally derived in [17].

**Lemma 7** ([17]). *Let $\Delta_k \triangleq \Delta_k(\epsilon) \to \infty$ be any diverging sequence satisfying $\Delta_k(\epsilon)/N_{k-\Delta_k(\epsilon)} \to 0$ and $\sum_{i=k-\Delta_k(\epsilon)}^{k-1} \eta_i \leq \epsilon$. Then, for all $\epsilon > 0$,*

$$\mathbb{P}[\phi(\theta^{(k)}) - \Phi(\theta^{(k)}, \lambda^{(k)}) > \epsilon] \leq (1 - q(\epsilon/2))^{\Delta_k} + \frac{32\sigma^2\Delta_k}{\epsilon^2 N_{k-\Delta_k}}.$$

*Furthermore, when there exists $\delta > 1$ such that $k^\delta/N_k \to 0$, one can construct a sequence $\Delta_k$ such that for any $b \in [0, 1)$, $\sum_{k \geq 1} b^{\Delta_k} < \infty$ and $\sum_{k \geq 1} \Delta_k/N_k < \infty$.*

Lemma 7 shows that as long as the observation noise is asymptotically mitigated, then one can guarantee convergence of $\Phi(\theta^{(k)}, \lambda^{(k)})$ to $\phi(\theta^{(k)})$. When $\eta_k = \Theta(1/\sqrt{k})$, we have $\Delta_k = \mathcal{O}(\sqrt{k})$. Below, we invoke Lemma 7 to prove our conclusion.

## C.1 Proof of (i): Asymptotic Convergence to a Stationary Point

Similar to the proof of Theorem 2, we have

$$\phi_\nu(\theta^{(k+1)}) \leq \phi(\widehat{\theta}^{(k)}) + \frac{1}{2\nu}\|\theta^{(k+1)} - \widehat{\theta}^{(k)}\|^2$$

$$= \phi(\widehat{\theta}^{(k)}) + \frac{1}{2\nu}\left\|\Pi\left[\theta^{(k)} - \eta_k\nabla_\theta\widehat{\Phi}_{N_k}(\theta^{(k)}, \lambda^{(k+1)})\right] - \Pi\left[\widehat{\theta}^{(k)}\right]\right\|^2$$

$$\leq \phi(\theta^{(k)}) + \frac{1}{2\nu}\left\|\theta^{(k)} - \widehat{\theta}^{(k)} - \eta_k\nabla_\theta\widehat{\Phi}_{N_k}(\theta^{(k)}, \lambda^{(k+1)})\right\|^2, \tag{47}$$

where the first inequality holds from the definition of the Moreau envelope, and the last inequality holds from the non-expansiveness of the projection operator. Further expanding the last term in (47), we can further upper bound $\phi_\nu(\theta^{(k+1)})$ by

$$\phi_\nu(\theta^{(k+1)}) \leq \phi(\theta^{(k)}) + \frac{1}{2\nu}\|\theta^{(k)} - \widehat{\theta}^{(k)}\|^2 + \frac{1}{2\nu}\|\eta_k\nabla_\theta\widehat{\Phi}_{N_k}(\theta^{(k)}, \lambda^{(k+1)})\|^2 +$$

$$+ \frac{1}{2\nu} \cdot 2\langle\theta^{(k)} - \widehat{\theta}^{(k)}, -\eta_k\nabla_\theta\widehat{\Phi}_{N_k}(\theta^{(k)}, \lambda^{(k+1)})\rangle$$

$$= \phi_\nu(\theta^{(k)}) + \frac{1}{2\nu}\|\eta_k\nabla_\theta\widehat{\Phi}_{N_k}(\theta^{(k)}, \lambda^{(k+1)})\|^2 + \frac{1}{\nu}\langle\theta^{(k)} - \widehat{\theta}^{(k)}, -\eta_k\nabla_\theta\widehat{\Phi}_{N_k}(\theta^{(k)}, \lambda^{(k+1)})\rangle$$

$$\leq \phi_\nu(\theta^{(k)}) + \frac{1}{2\nu}\eta_k^2\widehat{K}^2 + \frac{1}{\nu}\langle\theta^{(k)} - \widehat{\theta}^{(k)}, -\eta_k\nabla_\theta\widehat{\Phi}_{N_k}(\theta^{(k)}, \lambda^{(k+1)})\rangle, \tag{48}$$

where the first equality holds by the definition of the Moreau envelope and the construction of $\widehat{\theta}^{(k)}$, and the second inequality holds by upper bounding the gradient norm $\|\nabla_\theta\widehat{\Phi}_{N_k}(\theta^{(k-1)}, \lambda^{(k)})\|^2 \leq \widehat{K}^2$. Now, by the smoothness of $\phi$, we have

$$\phi(\widehat{\theta}^{(k)}) \geq \Phi(\widehat{\theta}^{(k)}, \lambda^{(k+1)})$$

$$\geq \Phi(\theta^{(k)}, \lambda^{(k+1)}) + \langle\nabla_\theta\Phi(\theta^{(k)}, \lambda^{(k+1)}), \widehat{\theta}^{(k)} - \theta^{(k)}\rangle - \frac{L}{2}\|\widehat{\theta}^{(k)} - \theta^{(k)}\|^2. \tag{49}$$

Hence, letting $\nu = 1/2L$, and plugging (49) into the unconditional expected version of (48), we immediately have

$$\mathbb{E}[\phi_{1/2L}(\theta^{(k+1)})] \le \mathbb{E}[\phi_{1/2L}(\theta^{(k)})] - 2L\eta_k \mathbb{E}\left[\phi(\theta^{(k)}) - \phi(\widehat{\theta}^{(k)}) - \frac{L}{2}\|\widehat{\theta}^{(k)} - \theta^{(k)}\|^2\right] +$$

$$+ 2L\eta_k \mathbb{E}\left[\phi(\theta^{(k)}) - \Phi(\theta^{(k)}, \lambda^{(k+1)})\right] + L\eta_k^2 \widehat{K}^2. \tag{50}$$

By the smoothness assumption on $\phi$, it is also a weakly convex function. In particular, $\phi(\theta) + L\|\theta - \theta^{(k)}\|^2$ is $L$ strongly convex. Hence,

$$\phi(\theta^{(k)}) - \phi(\widehat{\theta}^{(k)}) - \frac{L}{2}\|\widehat{\theta}^{(k)} - \theta^{(k)}\|^2 = \left(\phi(\theta^{(k)}) + L\|\theta^{(k)} - \theta^{(k)}\|^2\right) -$$

$$- \left(\phi(\widehat{\theta}^{(k)}) + L\|\widehat{\theta}^{(k)} - \theta^{(k)}\|^2\right) + \frac{L}{2}\|\widehat{\theta}^{(k)} - \theta^{(k)}\|^2$$

$$\ge L\|\widehat{\theta}^{(k)} - \theta^{(k)}\|^2 = \frac{1}{4L}\|\nabla\phi_{1/2L}(\theta^{(k)})\|^2. \tag{51}$$

Hence, plugging (51) into (50) immediately leads to

$$\mathbb{E}[\phi_{1/2L}(\theta^{(k+1)})] \le \mathbb{E}[\phi_{1/2L}(\theta^{(k)})] - \frac{\eta_k}{2}\mathbb{E}\|\nabla\phi_{1/2L}(\theta^{(k)})\|^2 +$$

$$+ 2L\eta_k \mathbb{E}\left[\phi(\theta^{(k)}) - \Phi(\theta^{(k)}, \lambda^{(k+1)})\right] + L\eta_k^2 \widehat{K}^2. \tag{52}$$

Finally, upon telescoping over $k$ from $k_0$ to $\infty$, and performing simple algebraic manipulations, we have

$$\frac{1}{2}\mathbb{E}\left[\sum_{k=k_0}^{\infty} \frac{\eta_k}{\sum_{t=k_0}^{\infty} \eta_t}\|\nabla\phi_{1/2L}(\theta^{(k)})\|^2\right] \le \frac{\mathbb{E}[\phi_{1/2L}(\theta^{(k_0)})] - \min_{\theta \in \Theta_h} \phi_{1/2L}(\theta)}{\sum_{k=k_0}^{\infty} \eta_k} +$$

$$+ 2L\frac{\sum_{k=k_0}^{\infty} \eta_k \mathbb{E}[\phi(\theta^{(k)}) - \Phi(\theta^{(k)}, \lambda^{(k+1)})]}{\sum_{k=k_0}^{\infty} \eta_k} +$$

$$+ L\widehat{K}^2 \frac{\sum_{k=k_0}^{\infty} \eta_k^2}{\sum_{k=k_0}^{\infty} \eta_k}. \tag{53}$$

By the assumption on the step size $\eta_k$, the first and third terms on the right-hand side vanish. By Lemma 3, we have $\Phi(\theta^{(k)}, \lambda^{(k+1)}) \to \phi(\theta^{(k)})$ in probability, and since $\Theta_h$ and $\Theta_u$ are compact and convex, $\phi(\theta^{(k)}) - \Phi(\theta^{(k)}, \lambda^{(k+1)})$ is bounded. In this case, convergence in probability implies convergence in expectation. Hence, by Cesàro's lemma, the second term on the right-hand side converges to the limit of $\mathbb{E}[\phi(\theta^{(k)}) - \Phi(\theta^{(k)}, \lambda^{(k+1)})]$, which is 0 as well.

Meanwhile, the left-hand side of (53) is lower bounded by $\frac{1}{2}\inf_{k \ge k_0}\mathbb{E}\|\nabla\phi_{1/2L}(\theta^{(k)})\|^2$, which is further lower bounded by $\frac{1}{2}\mathbb{E}\inf_{k \ge k_0}\|\nabla\phi_{1/2L}(\theta^{(k)})\|^2$, which converges to $\frac{1}{2}\mathbb{E}\liminf_{k \to \infty}\|\nabla\phi_{1/2L}(\theta^{(k)})\|^2$ as $k_0 \to \infty$. Hence, we have

$$\liminf_{k \to \infty} \|\nabla\phi_{1/2L}(\theta^{(k)})\| = 0 \tag{54}$$

in the mean square sense, and hence also in probability. Finally, when there exists $\delta > 1$ such that $k^\delta/N_k \to 0$, we have $\phi(\theta^{(k)}) - \Phi(\theta^{(k)}, \lambda^{(k+1)}) \to 0$ almost surely by invoking Lemma 7 together with union bound. In (53), this means that for any realization such that $\phi(\theta^{(k)}) - \Phi(\theta^{(k)}, \lambda^{(k+1)}) \to 0$, (54) holds true. Since such realizations happens with probability 1, (54) holds true almost surely. Thus, we have reached the conclusion for the convergence of $\|\nabla\phi_{1/2L}(\theta^{(T)})\|$.

We hence conclude the proof.

## C.2 Convergence Rate of GDRS for Stochastic Objective

Similar to the proof of Theorem 2, we have

$$\mathbb{E}[\phi(\theta^{(k)}) - \Phi(\theta^{(k)}, \lambda^{(k+1)})] \le D\left[(1 - q(\epsilon/2))^{\Delta_k} + \frac{32\sigma^2 \Delta_k}{\epsilon^2 N_{k-\Delta_k}}\right] + \epsilon. \tag{55}$$

Since $\eta_k = \Theta(1/\sqrt{k})$, we have $\Delta_k = \mathcal{O}(\sqrt{k})$. By the analysis of Theorem 2, we immediately see that the right-hand side of (55) is upper bounded by $\left(\frac{\log k}{\sqrt{k}}\right)^{\frac{1}{\delta_q+1}}$ when

$$\epsilon \triangleq \epsilon_k = \Theta\left(\left(\frac{\log k}{\sqrt{k}}\right)^{\frac{1}{\delta_q+1}}\right). \tag{56}$$

Notice that the second term on the right-hand side of (55) can be made arbitrarily small by increasing $N_{k-\Delta_k}$ for any given $k$. With simple manipulations, we see that the right-hand side of (55) is dominated by $\epsilon_k$ when we set

$$N_{k-\sqrt{k}} = \Omega\left(\frac{\sigma^2 k^{\frac{1}{2}\left(1+\frac{3}{\delta_q+1}\right)}}{\log^{\frac{3}{\delta_q+1}} k}\right). \tag{57}$$

The remainder of the proof follows from (43).