[Reviews · NeurIPS 2020]

Review 1

Summary and Contributions: The paper considers optimization problems where the objective a sum/expectation over a large number of micro tasks. Unlike typical applications of stochastic approximation methods, where the expectation is on the outside of the objective function, here it's inside, so approximating the objective by sampling micro tasks is biased. Prior work presented a dual formulation that trains a disciminator model. However, this is a troublesome saddle point problem. The authors instead show that the inner maximization in the saddle point objective can be replaced by simply sampling N times from a particular proposal distribution and taking the best. The key contribution of the paper is a very general theoretical result that this procedure will converge. Unfortunately, this general theoretical result is buried in a paper that is application focused. See my comment below.

Strengths: The paper provides a concrete, easy to implement algorithm for a type of saddle point problem that occurs often in modern ML. The method is backed by thorough theoretical convergence analysis.

Weaknesses: I have some core questions regarding the micro-macro setup below. These will need to be addressed in the author's response before I can fully evaluate the weaknesses of the paper.

Correctness: Unfortunately, I do not have the technical background to evaluate the correctness of the paper's theoretical statements. Overall, the empirical evaluation seems sound. My primary question is why the covid modeling task is simplified to the point where there are no features for each county. It's unclear to me that the modeling formalism that the paper puts forth is appropriate when there are such symmetries in the micro tasks.

Clarity: The methodological contribution of the paper is extremely general ("generative adversarial networks, adversarial machine learning, distributionally robust optimization"). However, the primary focus of the paper is on a very specific modeling setup. I would have liked to see the paper more focused on the theoretical result, with the particular model as a motivating example. I understand the desire to focus on a model that has relevance during this crazy pandemic time. However, the impact of the paper may be smaller because many readers that would potentially benefit from using Algorithm 1 may not read it.

Relation to Prior Work: Yes, the paper's method synthesizes ideas from a number of recent papers, and gives them appropriate credit.

Reproducibility: Yes

Additional Feedback: **update based on authors' response** Thanks for addressing my concerns regarding the computational complexity of your method vs. the baseline method. The answer is certainly non-trivial and you should re-work the exposition to make this clear. I'm fundamentally confused by the computational advantage of your proposed method (CSO). Since Biased SGD is offered as a baseline, I'm assuming we're in the regime where N is large, and thus evaluating the exact gradient of (2) is expensive (b.c. it's O(N)). However, when you move to the saddle point formulation, evaluating \Phi(lambda) is O(N). If we could tolerate a sum of O(N) micro-level terms, why didn't we just do gradient descent on (2) directly? In Sec 4.1 The aggregation ratio is very small (N=10). What about the baseline that uses no stochastic approximation and does direct gradient descent on theta? In Sec 4.3 the SIR model only has two parameters. You could do brute-force model training by simply trying values on a 2D grid. How does the solution you found compare to the optimum from this procedure? In Sec 4.3 I don't quite understand the setup resulting from throwing away all county-level features. Doesn't this introduce considerable symmetries in the model? How do the micro-level predictions differ across counties. If they are all the same, then the sum over N counties isn't meaningful.


Review 2

Summary and Contributions: This paper develops a framework to learn a two-stage model that predicts an outer random variable (referred to as "macroscopic" information) from knowledge of data from both the outer and an inner (referred to as "microscopic" information) == Post-rebuttal == The authors have addressed my criticism of the missing dimensionality factor in the updates. I hope they include this more explicitly in the final version

Strengths: I particularly appreciated: * Clarity and motivation of the model. The authors use very effective combination of examples and formulas to describe the model. The examples make the model easily understandable while the formalization (1), (2) is useful to ground the intuitions gained in the examples. * Algorithm description and convergence guarantees. I appreciated how the authors gave a clear description of the algorithm in form of pseudocode in "Algorithm 1" and provide a statement of the convergence guarantees. While their claims here should be nuanced (see next section), I appreciate the clarity of this part of the paper.

Weaknesses: My main criticism is their convergence analysis hides a dimensionality factor. For example, in L202 they claim "O(Log T/Sqrt(T))", which matches the best-known-rate of projected gradient method". However, this is not an apples-to-apples comparison, as the per-iteration cost of both methods is vastly different. In the method of Algorithm 1, the per-iteration cost depends on m, which in turn depends on the dimension.

Correctness: It is correct to the best of my knowledge.

Clarity: Paper is clearly written and easy to follow. I appreciate the diagrams made by the authors to convey the intuition behind the approach, which is complemented by the formulas expressing the model more rigorously.

Relation to Prior Work: I will comment mostly on the optimization part of the paper. Since I don't think algorithm [18] is well known in the machine learning community (at least I was not aware of it), I would have appreciated a more detailed discussions of the differences (if any) between Algorithm 1 and this work. Otherwise the amount of discussion with previous work seems appropriate to me.

Reproducibility: Yes

Additional Feedback: Please be precise when referring to sensible topics to avoid misunderstanding. For example, the title of Example two is "Formulating COVID with CSO". Clearly, not the whole disease is formulated with CSO. A more accurate title would be "United States COVID forecast with CSO". Please change accordingly


Review 3

Summary and Contributions: This paper propose a unified framework for learning from micro and macro data, which is different from existing methods. Experiments show some improvements over existing baselines.

Strengths: 1. The proposed framework is new; 2. Convergence rate of the optimization algorithm is rigorously analyzed; 3. Experiments validate the effectiveness of the proposed method; 4. The topic studied in this paper is relevant to the NeurIPS community.

Weaknesses: 1. Though the proposed formulation is new, the novelty is limited, which has been used a lot in other machine learning areas; Particularly, the minmax formulation is very popular, for instance, the objective of generative adversarial networks (GAN); The proposed optimization algorithm, GDRS, which is originally introduced in reference [18]; 2. Several hyper-parameters need to tune (such as \mu and m), and the efficiency is not studied. The hyperparameter m is set to different values on different tasks and this paper does not provide a principle way of setting it; In terms of efficiency, this paper does not study the training time and inference time to show the capability of scaling to large-scale problems. ======After reading author's feedback============ Thank the authors for addressing my concerns. I was convinced that the proposed method is novel and could be scaled up to large datasets.

Correctness: Yes, the claims, method, and empricial methodology are correct.

Clarity: Yes, this paper is generally well written.

Relation to Prior Work: Yes, existing realted work is clearly discussed.

Reproducibility: No

Additional Feedback:


Review 4

Summary and Contributions: The authors propose a new algorithm based on random search for the (apparently well known) SIR model formulation. The authors also provide guarantee of convergence to some stationary solution. Note: I'm not very familiar with this area.

Strengths: - The paper was well written. - Tackled an interesting problem (at least, framing the problem in relation to Covid-19 made it so).

Weaknesses: Given the datasets are relatable, it would've been nice to know exactly what .06% error rate corresponded to.

Correctness: AFAICT. I certainly did not understand the convergence proofs, but based on my limited knowledge O(logT/\sqrt(T)) I've seen/know for SGD for general convex functions without strong-convexity. But please do note that I'm not familiar with this area.

Clarity: Yes. I found the paper very accessible and providing appropriate references.

Relation to Prior Work: I'm not very familiar with this work, I certainly found the references helpful for me to understand this work.

Reproducibility: Yes

Additional Feedback: Post authors feedback Thanks for responding to my question. What I really meant was how many infections does .06 correspond to. In any case, it was a minor point .


Review 5

Summary and Contributions: The authors formulate the problem of "micro-macro learning" as a conditional stochastic optimization (CSO) problem. Then, they propose the algorithm "Gradient Descent with Random Search (GDRS)" to solve it. They show the rate of convergence of the algorithm as O(log(T)/sqrt(T)) outer-loop, where the complexity of the inner loop depends on the flatness of the loss function around the optimum.

Strengths: The paper and propose a *simple*, and generic, algorithm, that works in the non-convex setting. The assumptions for the rate of convergence are relatively mild. The requirements are 1) The function has bounded gradient, 2) The objective function to be smooth in its first argument, assuming we take the maximizer as the second argument, 3) There is a non-zero chance to find the optimum randomly and 4) the function that characterizes the probability to find the optimum randomly is fractional-differentiable when the tolerance goes to zero.

Weaknesses: When introducing the assumption, Assumption (4) is a bit elusive. First, there is no definition or reference to fractional calculus in the manuscript. I think this is worth mentioning the definition of the fractional derivative as well as precise examples. The same hold in the Theorem, where no concrete example where given, thus we have no idea how large m can be. Finally, again in the same spirit, a useful improvement could be a study on the impact of the distribution Q over q(epsilon) and delta_q.

Correctness: The paper seems correct, although I did not study in detail the proofs due to the tight schedule.

Clarity: The paper is very well written.

Relation to Prior Work: The algorithm seems novel, and the authors properly cite references to previous works.

Reproducibility: Yes

Additional Feedback: None.

[Author Response · NeurIPS 2020]

We thank all reviewers for their comments and acknowledgement of our contribution. All comments are *very* useful
and will be addressed in greater details in the revised version. Below we address each reviewer's comments separately.

**Response to Reviewer 1**:
**GDRS for GANs.** We thank the reviewer for recognizing our potential contribution to solving GANs, and will follow
up with more in-depth studies on the topic. Here, we reiterate that micro-macro modeling is also a very general
problem setting, as suggested by multiple examples in the introduction and numerical experiment sections.
**Symmetry in the micro tasks.** While the individual SIR model is not directly powered by micro-features such as
county population, we did provide asymmetry by aligning the model's output for each county with the first day of
a reported infection within that county (normalized by county population), which serves as an implicit feature. We
apologize and will eliminate such confusion in the revised version.
**Computation advantage of GDRS for large and small $N$.** The reviewer raised a very good point. Indeed, applying
full gradient descent to solve the empirical approximation of CSO,

$$\min_{\theta \in \Theta_h} \frac{1}{M} \sum_{i=1}^{M} \ell\left( \frac{1}{N} \sum_{j=1}^{N} h_\theta(\xi_i, x_{ij}), \bar{y}_i \right), \tag{1}$$

has the *same per-iteration computation* complexity as applying GDRS to solve the empirical approximation of the
minimax reformulation:

$$\min_{\theta \in \Theta_h} \max_{\lambda \in \Lambda_u} \Phi(\theta, \lambda) := \frac{1}{MN} \sum_{i=1}^{M} \sum_{j=1}^{N} \left\{ h_\theta(\xi_i, x_{ij}) u_\lambda(\xi_i, \bar{y}_i) - \ell^*(u_\lambda(\xi_i, \bar{y}_i)) \right\}. \tag{2}$$

However, using same amount of $MN$ samples and without assuming strong convexity conditions, the generalization
bound of (2) is of order $\mathcal{O}(1/\sqrt{MN})$, while the generalization bound of (1) is of order $\mathcal{O}(1/\sqrt{N} + 1/M)$ (Hu
et al.(2019)). This is an important reason why we decided to pursue the empirical approximation of the minimax
reformulation rather than the original CSO in the first place. We will add this clarification in the revised version.
**Grid search for SIR model.** Using fine-grained brute-force search will likely achieve an optimal solution; however,
the computational cost will be very high given the expensive cost of calling a differential equation solver for evaluating
each solution candidate. Our gradient-based method is much more efficient but only finds a stationary point. We are
afraid that such a comparison may not be fair.

**Response to Reviewer 2**:
**Convergence.** The reviewer is absolutely correct that the overall complexity depends on the dimension. Our claim
that "if $m = \log_{1-q(\epsilon)} \epsilon$, then GDRS converges to a neighborhood of the stationary point at rate $\mathcal{O}(\log T/\sqrt{T})$, which
matches the best-known-rate of projected gradient method for nonconvex minimization" is trying to emphasize that
when $m$ is sufficiently large, i.e., *with high per-iteration cost*, the *iteration complexity* of GDRS becomes the same as
projected gradient method. We will modify the sentence to avoid any confusion.

**Novelty/Relation to [18].** GDRS was proposed originally in [18] in 1983, but as reviewer pointed out, it is rarely
known to the machine learning community, which motivated us to speak it out for many potential applications, not
limited to macro-learning. Although we adopted the exact original form of GDRS, in this paper we extended the
asymptotic analysis of [18] to nonconvex-nonconcave objectives, and more importantly, we provided the first non-
asymptotic convergence analysis with explicit dependence on $m$ and other factors.

**Response to Reviewer 3**:
**Novelty/Relation to [18].** Please see the second point in response to Reviewer 2.
**Tuning hyperparameters.** We agree with the reviewer and will report tuning in the revised version. Here, $\nu$ only
appears in theoretical analysis and does not need tuning. In practice, we kept $m$ small to get fast run time.

**Response to Reviewer 4:**
**What does ".06" error rate correspond to?** ".06" is the *average error* between the observed infection numbers and
the estimated infection numbers over a period of 128 days on the testing counties.

**Response to Reviewer 5:**
**Elusive assumption and theorem statement.** We agree with the reviewer that fractional derivatives deserve more
detailed treatment in the main text, and we will provide examples illustrating the spirit of the theorem.
**Impact of the distribution $Q$.** This is a very good suggestion. We will provide analysis for specific distributions,
accompanied by numerical illustrations.

[Meta-Review · NeurIPS 2020]

The reviewers have supported the acceptance of this paper mostly due to the proposed optimization method and its analysis.